# A cross-sectional survey of false beliefs about COVID-19 and their association with vaccine hesitancy and uptake in the United States and China

Rachel E. Dinero[1,2], Mathlide Hou[1], Trijya Singh[3], Rong Chen[4], Brittany L. Kmush[5]*

**1** Department of Psychological and Brain Sciences, Colgate University, Hamilton, New York, United States of America, **2** Department of Psychology, Le Moyne College, Syracuse, New York, United States of America, **3** Department of Mathematics, Statistics, and Actuarial Science, Le Moyne College, Syracuse, New York, United States of America, **4** Department of Psychology, Dominican University of California, San Rafael, California, United States of America, **5** Department of Public Health, Syracuse University, Syracuse, New York, United States of America

* blkmush@syr.edu

## Abstract

Misinformation about COVID-19 vaccines fuel vaccine hesitancy and refusal. The prevalence of false beliefs contributes to differences in vaccine hesitancy across the US and China. We present cross sectional research assessing false beliefs about COVID-19 and the COVID-19 vaccine, and both vaccine/booster status and hesitancy in the US and China. We compared these variables across US (n=454) and Chinese (n=456) participants. Additionally, we use regression analyses to assess the relative association between false beliefs about COVID-19 and the COVID-19 vaccine and vaccine/booster hesitancy and uptake. The likelihood of receiving an initial vaccine was greater in China than the US (OR = 346.50, p<0.001), and false beliefs about COVID-19 were associated with a decreased likelihood of receiving an initial vaccine (OR = 0.82, p=0.05). The likelihood of receiving booster was just over three times greater in China than the US (OR = 3.45, p<0.001) and false beliefs about COVID-19 were not associated with the likelihood of receiving a booster (OR = 0.92, p=0.33). Vaccine hesitancy is more likely in the US than China ($e^B$=0.85, p<0.001) and false beliefs about COVID-19 are positively associated with vaccine hesitancy ($e^B$=1.20, p<0.01). There was significant interaction between country and false beliefs ($e^B$=0.93, p=0.02). Booster hesitancy is more likely in the US than China ($e^B$=0.95, p=0.04) and false beliefs about COVID-19 are positively associated with booster hesitancy ($e^B$=1.15, p<0.001). False beliefs about COVID-19 were associated with attitudes (vaccine hesitancy) but not behaviors (vaccine status). Additionally, we found that the strength of the association between false beliefs and attitudes differed between the two countries. Trusted resources to combat false beliefs, that are tailored to the local context, could help reduce vaccine hesitancy and increase vaccine uptake.

**Data availability statement:** All data, survey materials, and analysis syntax are available through OSF at https://osf.io/2hyba/?view_only=dacc14641c024b33aff7debd7f3ff404.

**Funding:** This research was funded by a Research Development Grant from Colgate University (RD). The funders had no role in study design, data collection and analysis, decision to publish, or preparation of the manuscript.

**Competing interests:** The authors have declared that no competing interests exist.

## Introduction

In 2020 the World Health Organization labeled the proliferation of misinformation during the COVID-19 an "infodemic" [1]. While the infodemic was global, the prevalence of misinformation spread differed across countries, and was significantly higher in the United States (US) than China [2]. This difference can be at least partially attributed to strict media regulations in China, which curtailed the spread of misinformation [3,4] and subsequent false beliefs (i.e., belief that misinformation is accurate) held about COVID-19 [5,6]. As misinformation about COVID-19 and the COVID-19 vaccine can fuel vaccine hesitancy (i.e., discomfort, uncertainty, or negative attitudes toward vaccination) and refusal (i.e., the behavior of not getting a vaccine) [7], it is possible that the prevalence of misinformation could contribute to differences in vaccine hesitancy across the US and China.

### Vaccine refusal and hesitancy in the US and China

Despite China's appreciably higher population (over 1.4 billion people [8] to the US's 340 million [8]), a significantly higher percent of its population is vaccinated. As of March 10, 2023, when the Johns Hopkins Coronavirus Research Center stopped collecting vaccine data by country, over 1.3 billion people in China had received at least one dose of the COVID-19 vaccine, reflecting approximately 92% of China's population [9]. In the US, just over 269 million people received at least one dose of the COVID-19 vaccine, approximately 82% of the US population [9]. While there are many factors that could account for the higher number and percent of vaccinated individuals in China (e.g., vaccine access, vaccine mandates, political structure, cultural orientation), vaccine hesitancy is a common factor driving vaccine refusal in both China and the US [10–12]. In a 2021 sample of 29,925 Chinese participants, 8% indicated unwillingness to vaccinate. In a 2021 survey of 2350 Chinese participants in occupations that were a high priority to receive the vaccine, 35% indicated some level of hesitancy toward the vaccine [10]. Less than 10%, however, reported being unwilling to vaccinate [10]. A 2021 systematic review of vaccine hesitancy in US found hesitancy rates that ranged from 8% to 88% depending on region and demographic group [11]. In a 2021 systematic review of worldwide vaccine acceptance rates, of the 33 countries studied, China had among the highest acceptance rates at 91%, while the US had among the lowest at 57% [12]. Although many factors may contribute to higher vaccine hesitancy in the US; here we focus specifically on the impact of false beliefs about COVID-19 and the vaccine that are fueled by the online spread of misinformation.

### Media regulations and misinformation spread in the US and China

In 1998, China's president, Xi Jinping, launched the Golden Shield Project aimed at increasing cybersecurity in China. The project, popularly known as the "Great Firewall of China" is an internet censorship system that blocks content that is considered against or inconsistent with government approved messaging [3]. Most international social media platforms are blocked in China (e.g., Facebook, Twitter, Instagram, YouTube, Pinterest, WhatsApp) [4]. Chinese citizens can access locally run social

media sources such as WeChat and Sina Weibo, which are strictly regulated by the government. Additionally, government agencies have created large amounts of government approved messaging that inundates these platforms [3]. By contrast, digital content in the US is mostly unregulated by the government, due to First Amendment protections. In fact, the US Supreme Court has struck down the majority of proposed internet and social media regulations, including regulations geared towards protecting minors [13].

These differences in regulations contributed to differences in the amount and type of information disseminated online during the COVID-19 pandemic. An analysis of 1.75 million Weibo messages posted about COVID-19 vaccines in 2020 indicated that posts about the vaccine were more likely to be positive, rather than negative, and express pride with regards to China's involvement in vaccine development [5]. One study comparing COVID-19 related posts on Weibo and Twitter in February 2020 found that over 50% of tweets and retweets on Twitter contained misinformation, while just 4% of posts and reposts on Weibo contained misinformation [6]. The most common pieces of misinformation shared on Twitter centered around inaccurate COVID prevention and treatment methods (e.g., flu vaccine, HIV medications), discrediting of official information, and the creation of COVID-19 by pharmaceutical companies to generate profit. The most reposted Weibo post inaccurately likened COVID-19 to the flu and common cold [6].

The dissemination of misinformation can result in false beliefs about COVID-19, which can, in turn, contribute to vaccine hesitancy. In a 2021 sample of 29,925 Chinese participants, endorsement of vaccine conspiracy beliefs (e.g., novel coronavirus is a hoax) was associated with vaccine hesitancy [10]. In the US, false beliefs about the COVID-19 vaccine (e.g., the COVID-19 pandemic is a hoax, the COVID vaccine contains a tracking microchip) have been identified as a key cause of vaccine hesitancy across multiple studies [14–16].

## The present research

Previous research indicates that the spread of misinformation online in the US is significantly higher than in China [5,6] and that vaccine hesitancy and refusal are higher in the US than in China [9]. The goal of the present research was to extend these finding to a direct assessment of false beliefs about COVID-19 and the vaccine at an individual level in both the US and China. Further, we assessed the extent to which these false beliefs are associated with vaccine hesitancy and refusal across both countries.

In this paper, we present cross sectional research assessing false beliefs about COVID-19 and the COVID-19 vaccine, and both vaccine/booster status and hesitancy in the US and China. We compared these variables across US and Chinese samples. Additionally, we use regression analyses to assess the relative association between false beliefs about COVID-19 and the COVID-19 vaccine and vaccine/booster hesitancy and uptake. We predicted that vaccine hesitancy and false beliefs about COVID-19 and the vaccine would be higher in the US than China. Further we predicted that false beliefs would predict vaccine hesitancy and refusal in both countries.

## Methods

### Study design, setting, and participants

Participants were recruited from the US and China through CloudResearch to complete an online survey hosted on Qualtrics. CloudResearch is an online research platform that manages participant recruitment and data quality for academic studies. Participants were compensated in accordance with the minimum wage standards for each country. Data was collected between March 1 and March 8, 2023.

### Variables

Participants completed an online survey assessing the predictor variables age, gender, socioeconomic status (SES), perceived COVID-19 vaccine access, and false beliefs about COVID-19, and outcome variables COVID-19 vaccine status, COVID-19 booster status, COVID-19 vaccine hesitancy, and COVID-19 booster hesitancy. The survey was created in

                                                                     

English and translated and back-translated by the second and third authors, respectively, who are fluent in English and Chinese [17]. The survey was also evaluated by a fluent Chinese speaker who was not affiliated with the project.

To assess false beliefs about COVID-19 and the vaccine, we generated a list of 16 pieces of misinformation that were identified from the literature as having been disseminated in the US and/or China [18–20] (e.g., "the COVID-19 virus in manmade", "vaccines only increase the profits of pharmaceutical companies and have no other purpose"). Participants indicated whether each piece of misinformation was "true", "false", or if they were "not sure". The endorsement of misinformation variable was the cumulative number of items endorsed as true.

Given that our data was collected in March 2023 and the COVID-19 booster doses became available in the US and China in 2021 [21,22], we assessed initial vaccine series status and hesitancy separately from booster status and hesitancy. For the purposes of this analysis, any COVID-19 vaccine dose given after 2021 was considered a booster, since that was the language used when those doses became available. Therefore, in the context of this study, a booster refers to any dose of the COVID-19 vaccine received after initial series of doses were available. As such, in this study, "vaccine" refers to the initial vaccine and "booster" refers to a subsequent annual vaccination dose. These were assessed separately to understand if vaccine attitudes and behavior changed from the initial vaccine release to subsequent annual doses, which could be administered independently of the initial vaccine dose. Vaccine and booster status were each assessed with a single item: have you ever received a COVID-19 vaccine/have you received a COVID-19 booster, with responses, yes, no, or I don't know. Vaccine hesitancy was assessed using six items (e.g., I am concerned about the safety of the COVID-19 vaccine) from the Oxford COVID-19 vaccine hesitancy scale [23,24], rated on a 6-point Likert scale with responses ranging from Disagree Strongly (1) to Agree Strongly (6). Items were averaged to form a vaccine hesitancy scale score. Booster hesitancy was assessed using 4 items designed for this study to specifically measure concerns about receiving a booster dose (i.e., "I worry about a serious adverse reaction after receiving the COVID-19 vaccine booster dose", "I believe it is safe to receive a COVID-19 vaccine booster dose", "Boosters are effective against SARS-CoV-2 variants", "I have a high level of fear associated with receiving the COVID-19 vaccine booster dose"). Items were averaged to form a booster hesitancy scale score. Vaccine access was assessed through a single item, "It would be relatively easy for me to get a COVID-19 vaccine if I wanted one", rated on the same 6-point Likert scale for vaccine hesitancy items.

## Research ethics and consent

This study was conducted in accordance with World Medical Association Declaration of Helsinki and approved by the second author's Institutional Review Board (Approval number ER_S23_03). Written informed consent was completed online by all participants.

## Planned analysis

Reliability for all scale scores (i.e., vaccine hesitancy, booster hesitancy, and perceived COVID-19 risk) was assessed using Cronbach's alpha. Differences in distributions of vaccine and booster status between Chinese and US samples were compared using Chi-squared tests of independence. Mean differences between Chinese and US samples in age, vaccine access, vaccine hesitancy, booster hesitancy, and false beliefs were assessed using independent samples t-tests. Similarly, we used independent samples -tests to assess differences in age, vaccine access, and false beliefs across vaccinated and unvaccinated, and boosted and unboosted.

We used various types of regression models to assess the association of country and COVID-19 false beliefs with vaccine status, booster status, vaccine hesitancy and booster hesitancy. For each model, we ran regression analysis on the sample of participants who responded to the outcome variable, and excluded participants who did respond to the outcome variable. We used logistic regression to model vaccine status and booster status as binary categorical outcome variables. For the vaccine status model, we used the Firth's penalized regression [25] to account for the quasi-complete separation

in the vaccine status variable resulting from all participants in the Chinese sample being vaccinated [26]. For both vaccine and booster status models, predicted probabilities for the logistic regression model were generated using the linear predictor (i.e., the log-odds) from the logistic regression model. Continuous covariates were held constant at their means and the categorical covariates were held constant at their reference category.

We used linear regression to model vaccine hesitancy and booster hesitancy as outcome variables, for both models we used the log transformation of the dependent variables (i.e., vaccine hesitancy and booster hesitancy) to account for skewed distributions in both variables. We used the Shapiro-Wilk's normality test to assess model residuals for normality, and the Breusch-Pagan test for diagnosing heteroscedasticity in the errors. Since the multiple linear regression models exhibited heteroscedasticity in the error terms, corrected robust standard errors were used to compute t-test statistics and p-value to account for this non-constant variation in the residuals. We also report exponentiated beta coefficients for these models for ease of interpretation. For both logistic and linear regression models, we first entered the control variables age, gender, SES, and vaccine access as Model 1. In Model 2 we added the main effects: country (US coded as 0, China coded as 1) and COVID-19 false beliefs. In Model 3 we added the interaction between country and false beliefs. All non-categorial independent variables (i.e., age, SES, misinformation, vaccine access) were entered as z-scores. Model fit for logistic regression was assessed using Akaike information criterion (AIC) and Bayesian information criterion (BIC). Model fit comparison was assessed using ANOVA. A two-sided P-value of < 0.05 was used to assess statistical significance. All analysis was completed in R version 4.4.1 [27].

## Results

There were 573 respondents from the US and 567 respondents from China. Participants who did not complete 80% of the survey ($N_{US}$ = 93, $N_{China}$ = 103), or did not pass all three attention checks ($N_{US}$ = 26, $N_{China}$ = 8) were removed from the dataset, resulting in 454 US participants (79% response rate) and 456 Chinese participants (80% response rate). Participant demographics for each sample are shown in Table 1. The vaccine hesitancy scale demonstrated adequate reliability (α = 0.88, 95% CI[0.87,0.89]). Vaccine hesitancy scores ranged from 1 to 6 out of a possible 6. The booster hesitancy scale demonstrated adequate reliability [α = 0.73, 95% CI[0.69,0.75]. Booster hesitancy scores ranged from 1 to 6 out of a possible 6. Means, standard deviations, and frequencies for demographic variables are shown in Table 1. Comparisons

**Table 1. Participant age, gender, and socioeconomic status (SES) for full sample, US and China subsamples, 2023.**

| | All Participants (n = 910) | US Sample (n = 454) | China Sample (n = 456) | US/China Comparisons |
|---|---|---|---|---|
| **Age** | | | | |
| Range | 18 – 86 | 18 – 86 | 18 – 70 | t(897) = 5.69 |
| Mean (Standard Deviation) | 35.96 (12.89) | 38.37 (15.31) | 33.56 (9.31) | p < 0.001 |
| **Gender** | %(n) | %(n) | %(n) | $X^2$ = 3.10 |
| Male | 34.8% (317) | 37.4% (170) | 32.2% (147) | p = 0.08 |
| Female | 63.3% (576) | 59.9% (272) | 66.7% (304) | |
| Nonbinary/no response | 1.9% (17) | 2.7% (12) | 1.1% (5) | |
| **SES** | | | | |
| Able to meet basic needs with ease and have a lot left over for extras | 27.3% (248) | 24.9% (113) | 29.6% (135) | $X^2$ = 58.25 |
| Able to meet basic needs with ease and have a little left over for extras | 48.2% (439) | 41.2% (187) | 55.3% (252) | p < 0.001 |
| Just able to meet basic needs | 18.5% (168) | 23.8% (108) | 13.2% (60) | |
| Do not have enough to meet basic needs | 5.6% (51) | 10.1% (46) | 1.1% (5) | |

Note: Gender and SES counts (n) are compared using $X^2$, while means (M) of age are compared using independent samples t-tests.

between the US and Chinese samples indicate that the US sample was, on average, older and lower on self-reported socioeconomic status.

For both vaccine status and booster status, there was a significant difference between China and the US in the distribution of "yes" and "no" responses (Table 2). Vaccine and booster hesitancy were higher in the US sample ($M = 3.30$, $M = 3.32$, respectively) as compared to China ($M = 2.45$, $M = 2.92$, respectively, p < 0.001). Despite the higher rates of vaccination in China, perceived vaccine access was higher in the US ($M = 4.61$) than China ($M = 4.37$). Additionally, the average number of endorsed misinformation items was higher in the US sample ($M = 3.16$) than the Chinese sample ($M = 1.86$).

Of the 16 false belief items, there was a significantly different response pattern across the US and China for 13 of the items (Table 3). The most frequently endorsed item by the US sample was "The COVID-19 pandemic leaked from a laboratory in Wuhan", with 40.1% of US participants endorsing this item as true. Only 5.9% of Chinese participants endorsed this item as true. False beliefs were generally lower for Chinese participants, with the exception of two items. In the Chinese sample, 23.9% of participants endorsed the item "The US military brought the COVID-19 pandemic to China", while only 12.1% of US participants endorsed this item (p < 0.001). The item "COVID-19 alone doesn't require hospitalization, severe illness only occurs in combination with other conditions" was the most frequently endorsed item by Chinese participants at 31.1%, compared to the US endorsement rate of 23.2% (p < 0.001).

Prior to running regression models, we assessed the association between all predictor variables (i.e., gender, country, age, vaccine access, false beliefs) and each outcome variable (i.e., vaccine status, booster status, vaccine hesitancy, booster hesitancy). As shown in Table 4, Chinese participants were more likely to report being vaccinated ($X^2 = 150.84$, p < 0.001) and boosted ($X^2 = 50.21$, p < 0.001). Additionally, Chinese participants reported lower levels of vaccine hesitancy ($M = 2.45$) compared to US participants ($M = 3.30$; $t = 9.90$, p < 0.001), and lower levels of booster hesitancy ($M = 2.92$) compared to US participants ($M = 3.32$; $t(906) = 5.65$, p < 0.001). COVID-19 false beliefs were more common among participants who reported not receiving an initial COVID-19 vaccination ($M = 3.27$, $SD = 3.54$) than participants who reported receiving an initial vaccine ($M = 2.37$, $SD = 3.03$; $t(891) = 3.08$, p < 0.001). The higher endorsement of false beliefs among participants who reported not receiving a COVID-19 booster ($M = 2.74$, $SD = 3.06$) as compared to participants who

**Table 2. Descriptive statistics for full sample and China and US subsamples with subsample comparisons, 2023.**

|  | Full Sample (*n* = 910) | US Sample (*n* = 454) | China Sample (*n* = 456) | US/China Comparisons |
|---|---|---|---|---|
| **Vaccine Status** | *n (%)* | *n (%)* | *n (%)* |  |
| Vaccinated | 760 (83.5%) | 318 (70.0%) | 442 (96.9%) | $X^2 = 150.84$, p < 0.001 |
| Unvaccinated | 133 (14.6%) | 133 (29.3%) | 0 (0%) |  |
| No response | 17 (1.9%) | 3 (0.7%) | 14 (3.1%) |  |
| **Booster Status** | *n (%)* | *n (%)* | *n (%)* |  |
| Boosted | 555 (61.0%) | 224 (49.3%) | 331 (72.6%) | $X^2 = 50.21$, p < 0.001 |
| Not Boosted | 347 (38.1%) | 225 (49.6%) | 122 (26.8%) |  |
| No response | 8 (0.9%) | 5 (1.1%) | 3 (0.6%) |  |
|  | **M(SD)** | **M(SD)** | **M(SD)** |  |
| Vaccine Access | 4.49 (1.35) | 4.61 (1.43) | 4.37 (1.26) | $t(907) = 2.74$, p = 0.006 |
| Vaccine Hesitancy | 2.87 (1.35) | 3.30 (1.53) | 2.45 (0.96) | $t(908) = 9.90$, *p* < 0.001 |
| Booster Hesitancy | 3.12 (1.11) | 3.32 (1.25) | 2.92 (0.90) | $t(906) = 5.65$, p < 0.001 |
| Misinformation | 2.51 (3.12) | 3.16 (3.65) | 1.86 (2.31) | $t(908) = 6.39$, p < 0.001 |

Note: For vaccine and booster status, country counts (n) are compared using $X^2$. Means of vaccine access, vaccine hesitancy, booster hesitancy, and misinformation are compared using independent samples t-tests. P-values for all tests are indicated in table.

**Table 3. Comparison of misinformation endorsement across US (N = 454) and China (N = 456), 2023.**

| | True | | False | | Not Sure | | |
|---|---|---|---|---|---|---|---|
| Item | US% (N) | China% (N) | US% (N) | China% (N) | US% (N) | China% (N) | $X^2$ p-value |
| The COVID-19 virus is a hoax | 11.9% (54) | 2.0% (9) | 67.8% (308) | 89.5% (408) | 20.3% (92) | 8.6% (39) | 67.55, < 0.001 |
| The COVID-19 virus is manmade | 32.5% (147) | 21.1% (96) | 29.4% (133) | 36.4% (166) | 38.2% (173) | 42.5% (194) | 15.54, < 0.001 |
| Large pharmaceutical companies artificially created the COVID-19 pandemic for profit | 15.9% (72) | 18.7% (85) | 54.3% (246) | 49.5% (225) | 29.8% (135) | 31.9% (145) | 2.37, 0.31 |
| Only the elderly can be infected with COVID-19 | 16.6% (75) | 2.0% (9) | 70.6% (320) | 92.8% (423) | 12.8% (58) | 5.3% (24) | 80.22, < 0.001 |
| The COVID-19 pandemic is spread through 5G/magnetic fields | 11.9% (54) | 4.4% (20) | 63.8% (289) | 78.5% (358) | 24.3% (110) | 17.1% (78) | 28.42, < 0.001 |
| The COVID-19 pandemic leaked from a laboratory in Wuhan | 40.1% (182) | 5.9% (27) | 24.7% (112) | 75.0% (342) | 35.2% (160) | 19.1% (87) | 253.04, <0.001 |
| The US military brought the COVID-19 pandemic to China | 12.1% (55) | 23.9% (109) | 61.0% (277) | 30.9% (141) | 26.9% (122) | 45.2% (206) | 83.54, < 0.001 |
| The coronavirus was created to force everyone to get vaccinated | 17.7% (80) | 5.5% (25) | 55.2% (250) | 70.2% (320) | 27.2% (123) | 24.3% (111) | 38.01, < 0.001 |
| The coronavirus vaccine will be used for mass sterilization | 16.4% (74) | 3.5% (16) | 55.4% (250) | 77.2% (352) | 28.2% (127) | 19.3% (88) | 61.71, < 0.001 |
| Antibody testing is a conspiracy to collect people's DNA | 13.9% (63) | 4.8% (22) | 54.6% (247) | 73.0% (333) | 31.4% (142) | 22.1% (101) | 39.43, < 0.001 |
| A booster shot for the coronavirus vaccine is not necessary | 23.8% (108) | 9.9% (45) | 44.4% (201) | 62.9% (286) | 31.8% (144) | 27.3% (124) | 42.27, < 0.001 |
| The coronavirus vaccine causes new variants of the coronavirus | 20.1% (91) | 14.7% (67) | 46.7% (211) | 62.5% (285) | 33.2% (150) | 22.8% (104) | 23.00, < 0.001 |
| The coronavirus vaccine destroys a person's immune system | 24.1% (109) | 27.6% (126) | 45.9% (208) | 48.0% (219) | 30.0% (136) | 24.3% (111) | 4.04, 0.13 |
| Adults and young children will not get seriously ill from the coronavirus | 14.3% (65) | 5.0% (23) | 60.5% (274) | 82.2% (375) | 25.2% (114) | 12.7% (58) | 53.99, < 0.001 |
| COVID-19 alone doesn't require hospitalization, severe illness only occurs in combination with other conditions | 23.2% (105) | 31.1% (142) | 51.7% (234) | 46.3% (211) | 25.2% (114) | 22.6% (103) | 7.28, 0.03 |
| Vaccines only increase the profits of pharmaceutical companies and have no other purpose | 21.9% (99) | 6.1% (28) | 50.1% (227) | 74.6% (340) | 28.0% (127) | 19.3% (88) | 69.28, < 0.001 |

Note: Response counts for misinformation items are compared using $X^2$. P-values for all tests are indicated in table.

reported receiving a booster (M = 2.37, SD = 3.17) was approaching significance (t(900) = 1.72, p = 0.08). COVID-19 false beliefs were positively correlated with vaccine hesitancy (r = 0.32, p < 0.001) and booster hesitancy (r = 0.34, p < 0.001).

Using a Firth's penalized likelihood logistic regression model to predict vaccine status (Table 4), the control model (Model 1) included age, gender, SES, and vaccine access (AIC = 635.69, BIC = 659.59). Adding in main effects of country and false beliefs (Model 2) significantly improved the model fit (AIC = 454.02, BIC = 487.48; $X^2$(2) = 178.25, p < 0.001). Model 3, adding in the interaction between country and false beliefs did not significantly increase the fit of the model (AIC = 455.43, BIC = 493.68; $X^2$(1) = 1.11, p = 0.29). Therefore, we use Model 2 for interpretation, which indicates that the likelihood of receiving an initial vaccine was greater in China than the US (OR = 346.50, p < 0.001), and false beliefs about COVID-19 was significantly associated with the likelihood of receiving an initial vaccine (OR = 0.82, p = 0.05).

**Table 4. Vaccine status, booster status, vaccine hesitancy, and booster hesitancy associations with age, gender, SES, vaccine access, country, and false beliefs about COVID-19.**

| | Vaccine Status | | | Booster Status | | | Vaccine Hesitancy (n=872) | | Booster Hesitancy (n=872) | |
|---|---|---|---|---|---|---|---|---|---|---|
| | Vaccinated (n=760) n | Not Vaccinated (n=133) n | p | Boosted (n=555) n | Not Boosted (n=347) n | p | M | p | M | p |
| **Gender** | | | | | | | | | | |
| Male | 272 | 38 | 0.12 | 207 | 110 | 0.11 | 2.86 | 0.94 | 3.10 | 0.76 |
| Female | 475 | 93 | | 339 | 230 | | 2.86 | | 3.13 | |
| Nonbinary/Missing | 13 | 2 | | 9 | 7 | | | | | |
| **Country** | | | | | | | | | | |
| China | 442 | 0 | < 0.001 | 331 | 122 | < 0.001 | 2.45 | < 0.001 | 2.92 | < 0.001 |
| United States | 318 | 133 | | 224 | 225 | | 3.30 | | 3.23 | |
| | M | M | p | M | M | p | r | p | r | p |
| Age | 36.05 | 35.36 | 0.57 | 37.06 | 34.37 | 0.002 | -0.02 | 0.57 | -0.08 | 0.02 |
| Vaccine Access | 4.55 | 4.15 | 0.001 | 4.60 | 4.30 | 0.001 | -0.15 | < 0.001 | -0.16 | < 0.001 |
| False Beliefs | 2.37 | 3.27 | 0.002 | 2.37 | 2.74 | 0.08 | 0.32 | < 0.001 | 0.34 | < 0.001 |

Note: For vaccine and booster status, gender and country counts (n) are compared using $X^2$, while means (M) of age, vaccine access and false beliefs are compared using independent samples t-tests. For vaccine hesitancy and booster hesitancy, means (M) for gender and country are compared using independent samples t-tests (males compared to females), while Pearson's r correlations (r) are reported for age, vaccine access, and false beliefs. P-values for all tests are indicated in table.

For booster status (Table 5, Fig 1), Model 1 included age, gender, SES, and vaccine access (AIC = 1074.72, BIC = 1098.47). Model 2, adding in main effect variables country and COVID-19 false beliefs significantly improved model fit (AIC = 1012.17, BIC = 1045.42; $X^2$ (2) = 66.55, p < 0.001). Model 3, adding in the interaction between country and false beliefs, did not significantly increase the fit of the model (AIC = 1013.63, BIC = 1051.63; $X^2$ (1) = 0.54, p = 0.46). Therefore, we use Model 2 (Table 5) for interpretation, which indicates that the likelihood of receiving a booster was just over three times greater in China than the US (OR = 3.45, p < 0.001) and false beliefs about COVID-19 were not associated with the likelihood of receiving a booster (OR = 0.92, p = 0.33).

Using robust standard errors as well as the log transformation of vaccine hesitancy (Table 5), Model 1 with control variables (i.e., age, gender, SES, and vaccine access) fit the data adequately (adjusted $R^2$ = 0.06, AIC = 1226.09, BIC = 1254.72). Model 2, adding in main effect variables country and COVID-19 false beliefs, fit the data adequately (adjusted $R^2$ = 0.21, AIC = 1083.32, BIC = 1121.49) and significantly improved the fit of the model (F(2) = 79.28, p < 0.001).

**Table 5. Logistic regression assessing likelihood of vaccine and booster status from country and COVID-19 false beliefs, controlling for age, gender, SES, and vaccine access (N=854).**

| | Vaccine Status (N=854) | | | Booster Status (N=854) | | |
|---|---|---|---|---|---|---|
| Predictor Variables | OR | [95%CI] | p | OR | [95%CI] | p |
| Age | 1.26 | [1.03,1.56] | 0.02 | 1.41 | [1.20,1.65] | < 0.001 |
| Gender: Male (vs female) | 1.59 | [0.99,2.57] | 0.05 | 1.46 | [1.06,2.02] | 0.02 |
| SES | 0.55 | [0.42,0.70] | < 0.001 | 0.65 | [0.54,0.79] | < 0.001 |
| Vaccine Access | 1.47 | [1.12,1.81] | < 0.001 | 1.30 | [1.12,1.52] | < 0.001 |
| Country: China (vs US) | 346.50 | [49,43789] | <0.001 | 3.45 | [2.49,4.81] | < 0.001 |
| False Beliefs | 0.82 | [0.67,1.00] | 0.05 | 0.92 | [0.79,1.08] | 0.33 |

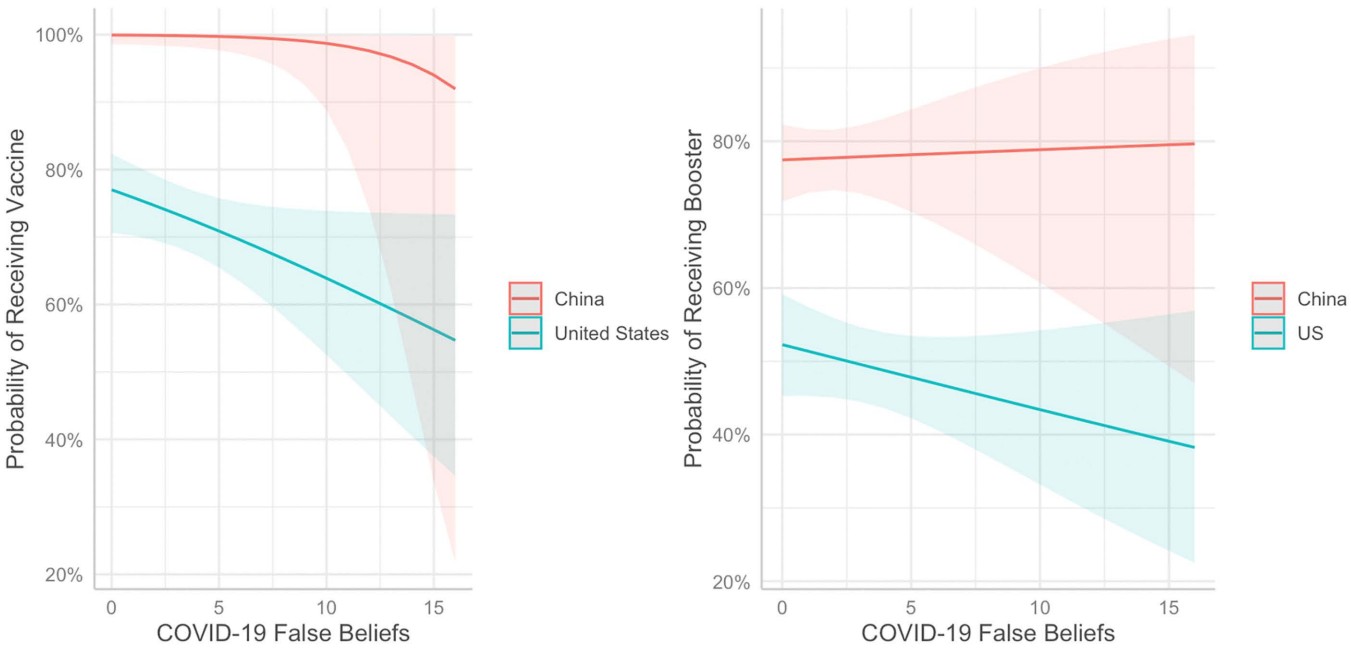

**Fig 1. Probability of receiving a COVID-19 vaccine and booster as a function of country and false beliefs about COVID-19 (N = 854).** Higher values indicate greater number of false beliefs endorsed (x-axis).

Adding in the interaction between country and false beliefs (Model 3) fit the data adequately (adjusted $R^2 = 0.21$, AIC = 1080.71, BIC = 1123.65) and significantly improved the fit of the model (F(1) = 4.58, p = 0.03). As such, we interpret Model 3, which indicates that when controlling for age, gender, SES, and vaccine access, vaccine hesitancy is more likely in the US than China ($e^B = 0.85$, p < 0.001) and false beliefs about COVID-19 are positively associated with vaccine hesitancy ($e^B = 1.20$, p < 0.01). Further, there was significant interaction between country and false beliefs ($e^B = 0.93$, p = 0.02) (Fig 2).

For booster hesitancy (Table 6, Fig 2), Model 1 fit the data adequately (adjusted $R^2 = 0.07$, AIC = 845.95, BIC = 874.58). Adding in main effect variables, country and COVID-19 false beliefs, fit the data adequately (adjusted $R^2 = 0.19$, AIC = 729.88, BIC = 768.05) and significantly improved the fit of the model (F(2) = 63.85, p < 0.001). Adding in interactions between country and false beliefs fit the data adequately (adjusted $R^2 = 0.19$, AIC = 731.38, BIC = 774.32), but did not significantly improve the fit of the model (F(1) = 0.50, p = 0.48). Therefore, we interpret Model 2. This model indicates that when controlling for age, gender, SES, and vaccine access, booster hesitancy is more likely in the US than China ($e^B = 0.95$, p = 0.04) and false beliefs about COVID-19 are positively associated with booster hesitancy ($e^B = 1.15$, p < 0.001) (Fig 2).

## Discussion

Consistent with previous research, vaccine and booster refusal, as well as vaccine and booster hesitancy were all higher in the US than China [5,6,9]. Further, our findings indicate that when controlling for age, gender, SES, and vaccine access, the probability of reporting receiving an initial dose of the COVID-19 vaccine was 346.5 times more likely in China than the US. Additionally, the likelihood of receiving a booster dose was 3.5 times more likely in China. Similarly, when controlling for age, gender, SES, and vaccine access, vaccine hesitancy was approximately 15% less in China than the US, and booster hesitancy was 5% less in China.

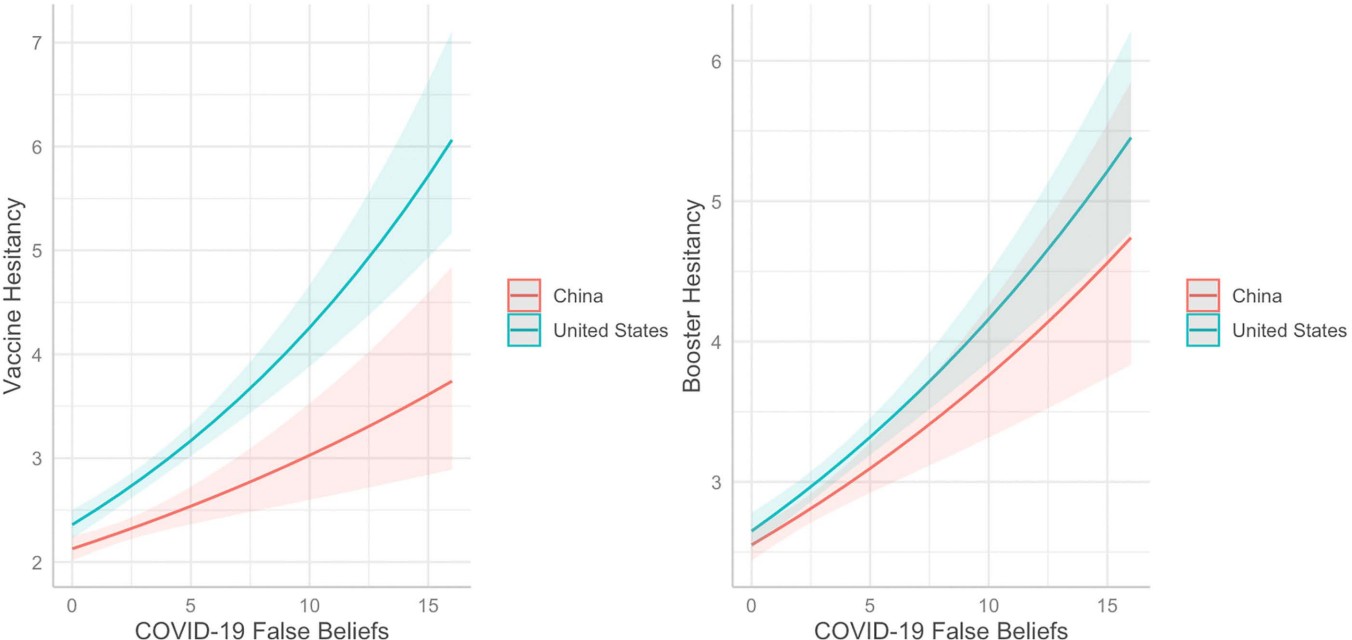

**Fig 2. Associations between country and false beliefs about COVID-19 and vaccine hesitancy, and booster hesitancy (N = 872).** Higher values indicate greater hesitancy (y-axis) and greater number of false beliefs endorsed (x-axis).

**Table 6. Linear regression assessing associations between vaccine and booster hesitancy with country and COVID-19 false beliefs, controlling for age, gender, SES, and vaccine access (N = 872).**

| Predictor Variables | Vaccine Hesitancy (N = 872) | | | Booster Hesitancy (N = 872) | | |
|---|---|---|---|---|---|---|
| | $e^B$ | [95%CI] | p | $e^B$ | [95%CI] | p |
| Age | 0.96 | [0.93,1.00] | 0.04 | 0.95 | [0.93,0.98] | 0.002 |
| Gender: Male (vs female) | 0.96 | [0.91,1.03] | 0.26 | 0.97 | [0.92,1.02] | 0.21 |
| SES | 1.12 | [1.07,1.16] | <0.001 | 1.08 | [1.04,1.11] | <0.001 |
| Vaccine Access | 0.91 | [0.88,0.94] | <0.001 | 0.92 | [0.90,0.95] | <0.001 |
| Country: China (vs US) | 0.85 | [0.80,0.91] | <0.001 | 0.95 | [0.90,1.00] | 0.04 |
| False Beliefs | 1.20 | [1.16,1.25] | <0.001 | – | – | – |
| Country*False Beliefs | 0.93 | [0.87,0.99] | 0.02 | – | – | – |

Note: The log transformation of the dependent variable was used in the model. The estimates presented in this table are the exponentiate of the model estimates. As such, they reflect the estimated change in the dependent variable for one unit change in each independent variable.

Novel findings from this research demonstrate that false beliefs were endorsed at a higher rate in the US than China. Of the 16 misinformation items shown to participants, 11 were more likely to be believed in the US. Only two misinformation items were more likely to be believed in China. In China, the most believed misinformation item was that COVID-19 only leads to severe illness and hospitalization in combination with other conditions. The only other item more commonly believed in China was that the US military brought the COVID-19 pandemic to China. Conversely, the most frequently believed misinformation item in the US was that the COVID-19 pandemic was leaked from a laboratory in Wuhan.

When controlling for age, gender, SES, and vaccine access, false beliefs about COVID-19 did decrease the likelihood of receiving an initial vaccine dose but were not associated with the likelihood of receiving a booster. False beliefs

PLOS Global Public Health

were also associated with increased vaccine and booster hesitancy. There was an interaction between country and false beliefs. This interaction indicated that while false beliefs were positively associated with vaccine hesitancy in both the US and China, this association was stronger in the US than in China.

## Implications

We found that false beliefs about COVID-19 were associated with attitudes (vaccine hesitancy) but not behaviors (vaccine status). Additionally, we found that location can play a role in the strength of the association between false beliefs and attitudes. People who are entrenched in anti-vaccine attitudes are unlikely to be persuaded to receive a vaccine, however, those who are more moderate in their vaccine hesitancy are often more open to vaccination, if their concerns can be addressed [28]. Our results demonstrate that false beliefs contribute to vaccine hesitancy, therefore, trusted resources with accurate information, via medical professionals, community leaders, or social media, could help decrease vaccine hesitancy. However, it is important to consider the local context and politics when developing and disseminating these resources.

## Limitations

As our data is cross-sectional, we cannot assume a causal path from exposure to misinformation to false beliefs about COVID-19 to vaccine hesitancy and refusal. Our data was collected through the participant recruitment platform CloudResearch, and our sample is unlikely to be fully representative of US and Chinese populations. Further, this data relies on self-report, which can be influenced by social desirability [29]. Social desirability while present cross-culturally, can manifest differently in individualist cultures, like the US, and collectivist cultures like China [30].

## Conclusions

COVID-19 vaccines contributed to reducing the morbidity and mortality associated with COVID-19 and helped hasten the return to normalcy. However, vaccine refusal and vaccine hesitancy have significantly limited the uptake of these vaccines. Vaccine refusal and hesitancy are not constat across geographic location. Here, we demonstrated that COVID vaccine and booster refusal are 346.5 and 3.5 times more likely in China than the US, respectively. Additionally, COVID vaccine and booster hesitancy was approximately 15% and 5% less in China than the US, respectively. Furthermore, false beliefs are endorsed at a higher rate in the US than China. Endorsing false beliefs about vaccines is a significant predictor of vaccine hesitancy, but not vaccine uptake in both countries. Trusted resources to combat false beliefs, that are tailored to the local context, could help reduce vaccine hesitancy. While international organizations such as the WHO provide general information about vaccines, local health departments and NGOs may be better positioned to provide context-specific information combatting false beliefs about vaccines, specifically designed to address the concerns of the population they serve.

## Author contributions

**Conceptualization:** Rachel E. Dinero, Mathlide Hou.

**Data curation:** Rachel E. Dinero, Mathlide Hou.

**Formal analysis:** Rachel E. Dinero, Mathlide Hou, Trijya Singh.

**Funding acquisition:** Rachel E. Dinero.

**Methodology:** Rachel E. Dinero, Mathlide Hou, Rong Chen.

**Project administration:** Rachel E. Dinero.

**Supervision:** Rachel E. Dinero, Brittany L. Kmush.

**Visualization:** Rachel E. Dinero.

**Writing – original draft:** Mathlide Hou, Trijya Singh.

**Writing – review & editing:** Rachel E. Dinero, Rong Chen, Brittany L. Kmush.

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
