## [Decision Letter · Decision Letter 0]

9 Sep 2025

PGPH-D-25-01987

A cross-sectional survey of false beliefs about COVID-19 and their association with vaccine hesitancy and uptake in the United States and China

Dear Dr. Kmush,

Thank you for submitting your manuscript to PLOS Global Public Health. After careful consideration, we feel that it has merit but does not fully meet PLOS Global Public Health’s publication criteria as it currently stands. Therefore, we invite you to submit a revised version of the manuscript that addresses the points raised during the review process.

The manuscript has been evaluated by two reviewers, and their comments are available below.

The reviewers have raised a number of concerns that need attention. In particular, they request additional information on methodological aspects of the study, and revisions to the statistical analyses.

Could you please revise the manuscript to carefully address the concerns raised?

We look forward to receiving your revised manuscript.

Kind regards,

Helen Howard

Staff Editor

Journal Requirements:

1. Please clarify all sources of funding (financial or material support) for your study. List the grants (with grant number) or organizations (with url) that supported your study, including funding received from your institution.

2. State the initials, alongside each funding source, of each author to receive each grant.

3. State what role the funders took in the study. If the funders had no role in your study, please state: “The funders had no role in study design, data collection and analysis, decision to publish, or preparation of the manuscript.”

4. If any authors received a salary from any of your funders, please state which authors and which funders.

2. Please send a completed 'Competing Interests' statement, including any COIs declared by your co-authors. If you have no competing interests to declare, please state "The authors have declared that no competing interests exist". Otherwise please declare all competing interests beginning with the statement "I have read the journal's policy and the authors of this manuscript have the following competing interests:"

3. Please provide separate figure files in .tif or .eps format.

4, If the reviewer comments include a recommendation to cite specific previously published works, please review and evaluate these publications to determine whether they are relevant and should be cited. There is no requirement to cite these works unless the editor has indicated otherwise.

Reviewers' comments:

Reviewer's Responses to Questions

**Comments to the Author**

1. Does this manuscript meet PLOS Global Public Health’s publication criteria?

Reviewer #1: Partly

Reviewer #2: Yes

2. Has the statistical analysis been performed appropriately and rigorously?

Reviewer #1: No

Reviewer #2: Yes

3. Have the authors made all data underlying the findings in their manuscript fully available (please refer to the Data Availability Statement at the start of the manuscript PDF file)?

Reviewer #1: Yes

Reviewer #2: Yes

4. Is the manuscript presented in an intelligible fashion and written in standard English?

Reviewer #1: Yes

Reviewer #2: Yes

Reviewer #1: General comment

The study provides an important insight on COVID-19 vaccine acceptance and uptake in two countries with diverse context. This study is of interest. But there are some issues related to data analysis and results presentation.

Introduction

L48-51 the objective of the study objective typically appears at the end of the introduction.

L60 I think it should be ‘higher proportion of vaccinated’.

L113-116 I think this section is similar to this section L123-125 I suggest to harmonize.

Methods

Kindly specify how you tested the normality of continuous variables such as age in the data analysis subsection.

Results

Kindly provide the response rate of this study.

The title 1 is not adequately written, kindly remove the redundant “and” in this title.

For P value I suggest to write the whole value for example instead of .08, write 0.08.

Kindly specify in footnote the meaning of t and X or X2 used in table 1. Is there a difference between X and X2 in this table? Not all reader might adequately identify that the number in parenthesis is the degree of freedom. I suggest the author to explain that in the footnote for more clarity. In summary specify in the footnote what is being presented in column US/China comparisons.

What do you mean by M and SD in table 1.

You could consider specifying after the name of the variable this ‘n (%)’ and adjust the content and position of number and percentage accordingly and then remove the repeating % and N=.

Usually for sample, the number of participants is noted n not N.

The interpretation of table 1 is sufficient. Kindly provide in one or two sentence a brief description of the results presented in table 1.

The total number and percentage are matching across variables nor for the total number of participants. If there were no response kindly specify that in the table as for table 2.

In table 2 the significance of results are not specified in the interpretation. Kindly adjust this.

Were all participants eligible for a booster dose as it is typically required for those who have already completed the initial number of required doses for the vaccine received.

Somebody which is not vaccinated should not be typically include as potential booster eligible individuals.

I think there might a bias in the interpretation of the booster hesitancy. How did you handle in this study the potential bias resulting from the partial vaccination against COVID-19 which might also result from hesitancy to received the second doses due concern about vaccine adverse effects or other reasons?

In the interpretation of table 3 kindly specify in the significance of results as your comparing two countries.

There is no need to repeat the ¨% after the value as this is specified in the heading of each column.

In table 4. It is known the best model is the lowest AIC’. why presenting multiple models with AIC instead of selecting and presenting the best model at the multivariate analysis.

Why not presenting the univariate regression analysis, near the multivariate analysis to enable the reader to identify the influence of the modelling or adjustment of potential covariates?

In table 4 I still have concerns about the validity of booster status assessment.

The age being a continuous variable its OR is not typically interpretable instead we should consider categorizing the age in other to interpret the OR. For continuous variable the sign and the value of the coefficient is used to interpret is influence in the probability of the dependent variable.

Kindly check the Y axis of figure 1. (probability).

How did you obtain the probability of receiving the vaccine for each participant in other the generate the figure 1.

What is the aim of measuring the interaction of some independent variables in these models?

The use of eb might not be appropriate in linear regression. The exponentiate is typically used in logistic regression modelling. The author could present the coefficient with its sign that is used to interpret the weight and the direction of influence of the independent variable as it increases or decreases in the model.

Other models present with low R2 (21% or 19%) instead the model with 85% best explain the dependent variable and the interpretation should emphasize in that model to avoid confusing the readers with multiple less reliable information.

Why is the sample size varying from table 4 to table 5. Kindly clarify this.

Specify the in Figure 2 in the Y and X axis vaccine hesitancy score or booster hesitancy score or covid-19 false beliefs score.

Discussion

Consider discussing the fact that despite the covid-19 false belief score is high in some participants in both countries, the US participants were consistently more hesitant or less likely to receive the vaccine.

Conclusion

The conclusion does nor report the main findings from the study. The comparison between countries is not reported. The mains independent variables explaining vaccine uptake or acceptance are not mentioned. Kindly suggest some strategy to combat false belief in these countries especially in US.

References

Kindly comply the referencing style ”PLOS ONE” of the journal.

Reviewer #2: 1. There is a need for a definition of terms. What is vaccine hesitancy? What is vaccine refusal? What are 'false beliefs'?

2. "While there are many factors." Use an alternative phrase here, as you have used this one earlier.

3. "We predicted that vaccine hesitancy and false beliefs about COVID-19 and the vaccine would be higher in the US than in China." This hypothesis seems to have a specified direction. Are you planning to conduct a one-tailed test?

4. Could the regulation of the internet in China also affect people's response to the survey?

5. How was the sample size determined? Inclusion and exclusion criteria? How do you ensure that there is no multiple response from a single individual? How much was the compensation?

6. Any linguistic qualification by these authors? This is important to ensure the meanings of the questions are preserved during translation. Fluency may not be enough to ensure accurate translation and backtranslation.

7. I think it will be helpful to describe how the variables were assessed better. State the 6-point Likert scale and what each point represents. Do this for all the variables. Also, be clear on the dependent and the independent variables in the methods section.

8. It is good practice to state the approval number of the protocol.

9. "A two-sided p-value =0.05." How will you use a two-sided p-value when your hypothesis was one-sided?

10. What is the response rate?

11. What is a geo-political location? Are you referring to the political climate in the US and China? Can you expatiate on this a bit more and relate it directly to your work?

12. Could the method of recruitment of participants be a limitation?

**Do you want your identity to be public for this peer review?** For information about this choice, including consent withdrawal, please see our Privacy Policy

Reviewer #1: **Yes:** Fabrice Zobel Lekeumo Cheuyem

Reviewer #2: **Yes:** Charles Olomofe

---

## [Decision Letter · Decision Letter 1]

4 Nov 2025

PGPH-D-25-01987R1

A cross-sectional survey of false beliefs about COVID-19 and their association with vaccine hesitancy and uptake in the United States and China

Dear Dr. Kmush,

Thank you for submitting your manuscript to PLOS Global Public Health. After careful consideration, we feel that it has merit but does not fully meet PLOS Global Public Health’s publication criteria as it currently stands. Therefore, we invite you to submit a revised version of the manuscript that addresses the points raised during the review process.

The revised manuscript has been reassessed and some concerns are still present. Please review the comments provided below and make the appropriate revisions to address the concerns raised, especially concerning the alignment of the conclusions.

We look forward to receiving your revised manuscript.

Kind regards,

Emma Campbell, Ph.D

Staff Editor

Journal Requirements:

Additional Editor Comments (if provided):

Reviewers' comments:

Reviewer's Responses to Questions

**Comments to the Author**

Reviewer #1: (No Response)

Reviewer #2: All comments have been addressed

publication criteria?

Reviewer #1: Yes

Reviewer #2: Yes

3. Has the statistical analysis been performed appropriately and rigorously?

Reviewer #1: Yes

Reviewer #2: Yes

4. Have the authors made all data underlying the findings in their manuscript fully available (please refer to the Data Availability Statement at the start of the manuscript PDF file)?

Reviewer #1: Yes

Reviewer #2: Yes

5. Is the manuscript presented in an intelligible fashion and written in standard English?

Reviewer #1: Yes

Reviewer #2: Yes

Reviewer #1: General comment

The authors have done a great revision in their manuscript to improve its quality. However, there are some points that were not adequately addressed. Find below my few remaining comments.

Results

I am not convinced with the justification of not performing any simple test to assess the normality and only assume based on the sample size. You can specify that as a limitation of this study.

I might not have been able to see the response rate in the first paragraph but only number.

For P value I suggest to write the whole value for example instead of .08, write 0.08. this was not applied as recommended.

Somebody which is not vaccinated should not be typically include as potential booster eligible individuals.

I think the author should acknowledge the fact that the analysis did not take into consideration the fact that booster hesitancy included also those who have been previously hesitant to the initial shot of vaccine.

I agree with this explanation but it should be clearly specified in the methodology. “1) Linear Prediction: For each value of the predictors specified in the terms argument (misinfo and countryR), the function calculated the linear predictor (log-odds) using the coefficient estimates from the Firth-penalized logistic regression model (logistf). All other covariates in the model (age, gender, SES, vaccine access) were held constant at their mean (for continuous variables) or reference level (for categorical variables).”

Specify the in Figure 2 in the Y and X axis vaccine hesitancy score or booster hesitancy score or covid-19 false beliefs score. This should be considered to enable to reader understand the meaning of the variable.

Conclusion

The conclusion does not report the main findings from the study. The comparison between countries is not reported. The mains independent variables explaining vaccine uptake or acceptance are not mentioned. Kindly suggest some strategy to combat false belief in these countries especially in US. I don’t agree with the justification provided by author for not clearly aligning their conclusion with some standards.

Reviewer #2: Everything looks good

**Do you want your identity to be public for this peer review?** For information about this choice, including consent withdrawal, please see our Privacy Policy

Reviewer #1: **Yes:** Fabrice Zobel Cheuyem Lekeumo

Reviewer #2: **Yes:** Charles Olomofe

---

## [Decision Letter · Decision Letter 2]

8 Jan 2026

PGPH-D-25-01987R2

A cross-sectional survey of false beliefs about COVID-19 and their association with vaccine hesitancy and uptake in the United States and China

Dear Dr. Kmush,

Thank you for submitting your manuscript to PLOS Global Public Health. After careful consideration, we feel that it has merit but does not fully meet PLOS Global Public Health’s publication criteria as it currently stands. Therefore, we invite you to submit a revised version of the manuscript that addresses the points raised during the review process.

The reviewer has identified some final copyediting revisions which are required before we can proceed with acceptance of your manuscript. Please carefully check your manuscript to make all necessary changes and updates in your resubmission.

We look forward to receiving your revised manuscript.

Kind regards,

Jennifer Tucker, PhD

Staff Editor

Journal Requirements:

Additional Editor Comments (if provided):

Reviewers' comments:

Reviewer's Responses to Questions

**Comments to the Author**

Reviewer #1: All comments have been addressed

publication criteria?

Reviewer #1: Yes

3. Has the statistical analysis been performed appropriately and rigorously?

Reviewer #1: Yes

4. Have the authors made all data underlying the findings in their manuscript fully available (please refer to the Data Availability Statement at the start of the manuscript PDF file)?

Reviewer #1: Yes

5. Is the manuscript presented in an intelligible fashion and written in standard English?

Reviewer #1: Yes

Reviewer #1: Thank you for the revisions and congratulations.

There is still formatting style issue in this manuscript in writing decimal value like .21 instead of 0.21 or < .001 instead of < 0.001. If this can be addressed by the copyediting service of the journal, it is ok for me. If not, I suggest to author to do it as

The word score is missing in the X and the Y axis of figure 2. This can be added in the footnote of this figure.

**Do you want your identity to be public for this peer review?** For information about this choice, including consent withdrawal, please see our Privacy Policy

Reviewer #1: **Yes:** Fabrice Zobel Lekeumo Cheuyem

---

## [Editor Report · Decision Letter 3]

20 Jan 2026

A cross-sectional survey of false beliefs about COVID-19 and their association with vaccine hesitancy and uptake in the United States and China

PGPH-D-25-01987R3

Dear Dr. Kmush,

We are pleased to inform you that your manuscript 'A cross-sectional survey of false beliefs about COVID-19 and their association with vaccine hesitancy and uptake in the United States and China' has been provisionally accepted for publication in PLOS Global Public Health.

Best regards,

Julia Robinson

Executive Editor